# Physical state of water controls friction of gabbro-built faults

Wei Feng[1,2], Lu Yao [2] ✉, Chiara Cornelio [3], Rodrigo Gomila [1], Shengli Ma[2], Chaoqun Yang[4], Luigi Germinario [1], Claudio Mazzoli [1] & Giulio Di Toro [1,3] ✉

Earthquakes often occur along faults in the presence of hot, pressurized water. Here we exploit a new experimental device to study friction in gabbro faults with water in vapor, liquid and supercritical states (water temperature and pressure up to 400 °C and 30 MPa, respectively). The experimental faults are sheared over slip velocities from 1 μm/s to 100 mm/s and slip distances up to 3 m (seismic deformation conditions). Here, we show with water in the vapor state, fault friction decreases with increasing slip distance and velocity. However, when water is in the liquid or supercritical state, friction decreases with slip distance, regardless of slip velocity. We propose that the formation of weak minerals, the chemical bonding properties of water and (elasto)hydrodynamic lubrication may explain the weakening behavior of the experimental faults. In nature, the transition of water from liquid or supercritical to vapor state can cause an abrupt increase in fault friction that can stop or delay the nucleation phase of an earthquake.

Seismological, geophysical, and deep borehole data plus field observations of exhumed faults are consistent with the hypothesis that crustal earthquakes often originate and propagate in the presence of hot and pressurized (i.e., hydrothermal) fluids[1–3]. Moreover, experimental studies indicate that hydrothermal fluids (usually water) govern the seismic cycle by means of a plethora of physical and chemical processes[4–7] (pore pressure variations, diffusive mass transfer, mineral precipitation, sub-critical crack growth, etc.), which determine, for instance, fault frictional instability or its healing and sealing. However, the role of the physical state (liquid, vapor, and supercritical) of water in the seismic cycle remains unclear, especially in relation to the safe exploitation of deep geothermal reservoirs[8]. Rock friction experiments performed over a broad range of slip distances (up to meters) and slip velocities (from a few μm/s, or sub-seismic, up to m/s, or seismic) can contribute to the understanding of the seismic cycle[9–19]. So far, because of technical challenges, rock friction experiments[15,16,20–25] under hydrothermal conditions have been limited to low slip velocities (≤-100 μm/s) and short slip distances (≤-80 mm). As a consequence, the frictional properties of fault rocks at seismogenic conditions in the presence of water under different physical states are still poorly understood.

Here we exploit a new experimental setup to investigate the effect of the physical state of water on fault friction in the seismic cycle. In the case of gabbro, a common crustal rock, liquid (but at high temperature and pressure) and supercritical water result in a large decrease in fault friction regardless of the imposed fault slip velocity, while in the case of vapor fault friction decreases with slip velocity. We propose three possible mechanisms which may contribute to the reduction of fault friction during shear: (1) the formation of clay minerals along slickensides due to water-rock interaction; (2) the H–O–H bonding properties of water at hydrothermal conditions; (3) viscous "slurry-like" lubrication due to the presence of water and rock powders produced during slip.

## Results

### Experimental protocol

We performed 25 rock friction experiments on cylindrical samples (28 mm external diameter) of gabbro in a rotary shear machine[26] equipped with an on-purpose designed hydrothermal vessel (for details on experimental setup, rock composition, list of experiments and data reproducibility see "Methods", Supplementary Figs. 1–3 and

[1]Dipartimento di Geoscienze, Università degli Studi di Padova, Padua, Italy. [2]State Key Laboratory of Earthquake Dynamics, Institute of Geology, China Earthquake Administration, Beijing, China. [3]Sezione Roma 1, Istituto Nazionale di Geofisica e Vulcanologia, Rome, Italy. [4]State Key Laboratory of Oil and Gas Reservoir Geology and Exploitation, Chengdu University of Technology, Chengdu, China. ✉e-mail: luyao@ies.ac.cn; giulio.ditoro@unipd.it

Supplementary Tables 1, 2). Two gabbro cylinders were sheared for slip distances up to 3 m at constant sub-seismic to seismic slip velocities ($V$ from 1 μm/s to 0.1 m/s) under drained "hydrothermal conditions" in the presence of pressurized distilled water in liquid (temperature $T = 300$ °C, pore fluid pressure $P_f = 10$ MPa), vapor ($T = 400$ °C, $P_f = 10$ MPa) and supercritical ($T = 400$ °C, $P_f = 30$ MPa) states. In all the experiments the imposed effective normal stress ($\sigma_n^{eff} = \sigma_n - \alpha\, P_f$ assuming $\alpha=1$, where $\sigma_n$ is normal stress) was 10 MPa. Fault strength[1] is reported as friction coefficient $\mu = \tau/\sigma_n^{eff}$, where $\tau$ is shear stress. The samples were recovered after the experiments for microanalysis (see "Methods") with Field Emission Gun Scanning Electron Microscope (FESEM), X-Ray powder diffraction (XRPD), Micro-Raman spectroscopy, and non-contact optical profilometer (OP) techniques.

### Mechanical results

In the experiments, the friction coefficient of gabbro evolved with slip distance, slip velocity and physical state of water as follows (Fig. 1):

Water in the liquid state. Regardless of the imposed slip velocity, slip-weakening behavior was observed. The friction coefficient of gabbro increased over a slip distance of 10–30 mm from an initial static friction coefficient $\mu_s = 0.61$–0.80 to the peak one $\mu_p = 0.75\pm0.10$, and then decayed over a slip distance of ~200–500 mm to the steady-state $\mu_{ss} = 0.20$–0.35 (Fig. 1a). Despite the similar trends of slip-weakening, the values of $\mu_s$, $\mu_p$ and $\mu_{ss}$, though with some scatter, decreased with increasing slip velocity (Fig. 1b).

Water in the supercritical state. Regardless of the imposed slip velocity and similar to experiments performed with water in the liquid state, the friction coefficient of gabbro increased over a slip distance of 10–280 mm from $\mu_s = 0.5$–0.58 to $\mu_p = 0.6 \pm 0.10$, and then decayed over a slip distance of ~300–500 mm to $\mu_{ss} = 0.2$–0.3 (Fig. 1c). The $\mu_s$, $\mu_p$, and $\mu_{ss}$ values were almost independent of slip velocity (Fig. 1d).

Water in the vapor state. Differently from the experiments with water in liquid and supercritical states, the friction behavior of gabbro was velocity-dependent (Fig. 1e). In particular, for $0.01 \leq V \leq 1$ mm/s, the friction coefficient remained high: from $\mu_s = 0.75 \pm 0.05$ it

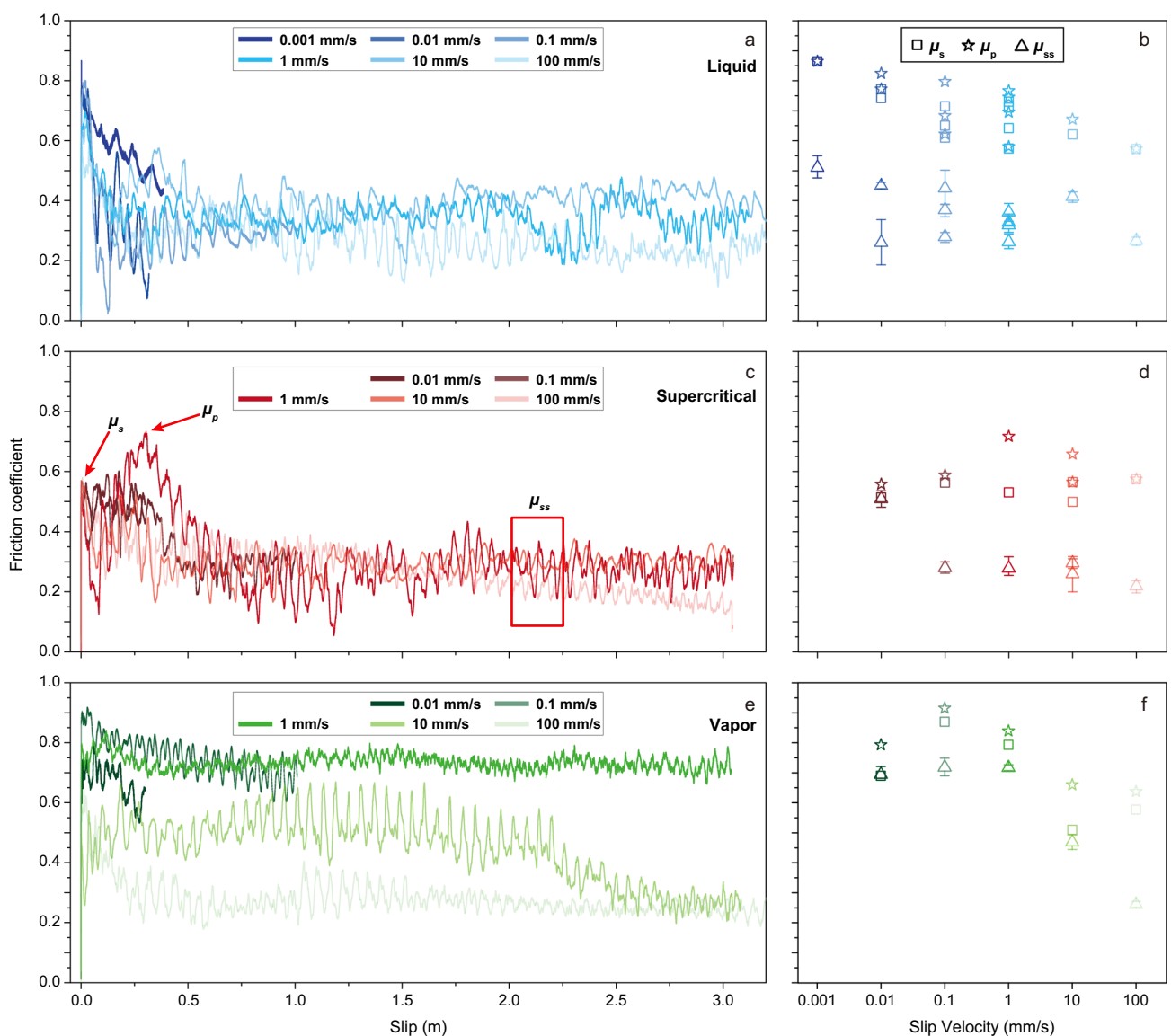

**Fig. 1 | Friction coefficient of gabbro rocks under hydrothermal conditions.** The gabbro cylinders were sheared up to 3 m of slip distance under a constant effective normal stress of 10 MPa. Left side graphs: evolution of friction coefficient with slip distance. Slip velocities ranged from 0.001 mm/s to 100 mm/s (colors coded). Right side graphs: static ($\mu_s$, open squares), peak ($\mu_p$ open pentagrams) and steady state ($\mu_{ss}$ open triangles; error bar represents the standard deviation of the data) friction coefficients with slip velocity. **a, b** Water in liquid state (temperature ($T$) = 300 °C, pore pressure ($P_f$) = 10 MPa). **c, d** Water in supercritical state ($T = 400$ °C, $P_f = 30$ MPa). **e, f.** Water in vapor state ($T = 400$ °C, $P_f = 10$ MPa).

increased to $\mu_p = 0.81-0.92$ and then decreased to $\mu_{ss} = 0.67-0.72$ (Fig. 1f). Instead, for $V = 10$ mm/s, the friction increased from $\mu_s = 0.51$ to $\mu_p = 0.66$ and then first decayed to $0.47 \pm 0.05$ over a slip distance of 2200 mm and then further decreased to 0.2 till the end of slip. At $V = 100$ mm/s, the friction coefficient decayed from $\mu_s = 0.57$ and $\mu_p = 0.64$ to $\mu_{ss} = 0.26$ after ca. 250 mm of slip. The $\mu_s$, $\mu_p$ and $\mu_{ss}$ values evolved from velocity-neutral to velocity weakening with increasing slip velocity (Fig. 1f).

The mechanical data show that the physical state of water impacts the friction properties of the experimental fault to a much greater extent than previously thought. For instance, in experiments performed with tri-axial machines (i.e., slip distance <1 mm, slip velocity <0.1 μm/s) on quartz gouges, the decrease of friction coefficient at the water transition from the liquid to the vapor state was limited to ~0.05[27]. Instead, we measured an increase in friction coefficient at this transition for larger slip distances and velocities (Fig. 1b, f).

### Micro-analytical results

To identify the possible deformation mechanisms responsible for these large measured differences in the frictional behavior of gabbro rocks, the samples were recovered for mineralogical and microstructural investigations. Unfortunately, part of the non-cohesive powder produced during shearing in the slip zone might have been flushed away during fluid ejection and sample unloading. However, regardless of the imposed slip velocity, the slip surfaces of the faults that were whitish in color before the experiments acquired a dark green color in the experiments with liquid and supercritical water and medium green color in the experiments with water in the vapor state (left panel of Fig. 2a–c). After the experiments, the slip surfaces were ultra-polished and lineated (i.e., striae parallel to the slip vector, see Supplementary Fig. 4) and, based on optical profilometer measurements, had a micro-roughness with a root mean square (RMS) of ~3.5–7 μm (middle panel in Fig. 2a–c) (for starting surface see Supplementary Fig. 5). The slip zones beneath the slip surfaces were <30 μm thick (inset in Fig. 2b for the experiment performed with supercritical state water) and made of ultrafine powders with grain size down to few nm (right panel in Fig. 2a–c). In particular, the powders from the experiments conducted with water in the supercritical state included ultra-fine (~200 nm long and ~30 nm thick) newly formed clay-like minerals (yellow arrow in the right panel of Fig. 2b). However, their size was too small for FESEM-elemental analysis. XRPD analysis performed directly on the slip surfaces (Fig. 2d) and on the powders (~10 mg, after ethylene glycol treatment, Supplementary Fig. 6) recovered from the slip zone revealed the presence of a broad peak at 8° in the spectra of the gabbro sheared with water in the liquid or supercritical state. This broad peak, typical of smectite and chlorite[28], was absent in the XRD spectra of the surfaces of the starting (not sheared) gabbro and of the slip surfaces recovered after the experiments with water in the vapor state. Based on XRD semi-quantitative analysis, the abundance of these newly formed minerals is estimated to be less than 5 wt.%. Similarly, a peak at a wavenumber of 1595–1600 cm$^{-1}$ was present only in the micro-Raman spectra of the ultra-polished slip surfaces recovered from the experiments performed with water in the liquid and supercritical states (Fig. 2e). This wavenumber corresponds to the H–O–H bending vibrational mode of $H_2O$[29] and is possibly associated with the water absorbed in the slip surfaces or immediately beneath (the micro-Raman exciting signal can penetrate <1 μm). In the micro-Raman spectra map ($600 \times 600$ μm$^2$) of the slip surfaces, the signal of the H–O–H bending overlaps with the location of the striae (Fig. 2f and Supplementary Fig. 7). This implies that the formation of this particular H-O-H bond is associated with the development of high-strain zones.

In conclusion, the microstructural and microanalytical analysis confirm the formation of an ultra-polished slip surface topping a few tens of micrometers (inset in Fig. 2b) thick slip zone in all the experiments. However, only in the experiments performed with liquid or supercritical water (i.e., when, regardless of the imposed slip velocities, the lowest values of the friction coefficient were measured; Fig. 1) the slip zone is enriched in H-O-H bonds water and includes newly-formed water-bearing minerals (Fig. 2).

## Discussion

In this study, we imposed with the external furnace and pressuring fluid pump, temperature and pressure conditions to obtain a constant physical state of water (liquid, supercritical, and vapor) inside the hydrothermal vessel during the experiments. However, frictional heating can lead to an increase in the temperature in the slipping zone which may determine the transition of the physical state of water. Numerical modeling (see "Methods") shows that the temperature increase in the slipping zone required for the transition of water from liquid to vapor state is achieved in experiments performed at $V \geq 10$ mm/s (Supplementary Fig. 8). This result is supported by the observation that $\mu_{ss}$ is similar between the experiments in liquid and vapor water at $V > 10$ mm/s (Fig. 1b, f). Whereas in the experiments performed at $V \leq 1$ mm/s, temperature increase in the slipping zone is negligible and the physical state of water should not change.

Thousands of experiments performed in the last 25 years aimed at approaching or reproducing seismic deformation conditions (e.g., $V \geq 0.1$ m/s) measured low values of $\mu_{ss}$ independently of rock composition and of the pressure of water[30]. This so-called dynamic weakening is due to the activation of several temperature and possibly grain-size dependent weakening mechanisms[30–36]. However, these previous observations are at odds with the experimental data presented in Fig. 1. For instance, in previous experiments performed at room temperature on gabbro either under room humidity[35] or in the presence of pressurized liquid water, $\mu_{ss}$ decreases only for $V \geq 10$ mm/s[5,37], not for $V \geq 0.01$ mm/s when pressurized (hot) water is in liquid and supercritical states (Fig. 1a, b). The weakening measured at $V = 100$ mm/s could be attributed to several established thermal weakening mechanisms (e.g., flash heating[36,38], thermal pressurization[31,39], frictional melting[40], etc.). We assess the weakening effect of thermal pressurization as an example via numerical modeling[39]. The results show that only at $V = 100$ mm/s pore pressure increase in the slipping zone could potentially contribute to the observed weakening (see Supplementary Fig. 9). Flash heating is also likely to occur at $V \geq 100$ mm/s in accordance with previous studies[36,38]. However, the weakening behavior observed at slip velocities $\leq 10$ mm/s cannot be explained by these mechanisms.

Alternatively, low values of $\mu_{ss}$ were measured in experiments performed on particular clays and phyllosilicates (e.g., saponite, Na-smectites, talc) gouges sheared at sub-seismic slip velocities ($V \leq 10$ mm/s)[41–43]. Also mixtures of these particular clays/phyllosilicates with "hard" minerals (e.g., quartz, calcite, dolomite) may result in a bulk low friction coefficient if the "weak" minerals form a continuous layer[44]. As a consequence, the low friction coefficient measured at sub-seismic slip rates in the experiments performed with water in liquid or supercritical state could be related to the formation of minerals with low frictional strength. Their formation, due to fluid-rock interaction, would be boosted by the increased mineral reactivity due to tribochemical effects in the slip zone because of grain size reduction and the high temperature of the fluid. In fact, newly-formed clay-like minerals are present in the recovered slip zones (Fig. 2a–c). Although their abundance is very low (<5 wt.%), and clays are not arranged to form a continuous layer, we cannot rule out that clay-rich parts of the slip zones were actually lost during sample recovery. Moreover, the coincidence of the distribution of H-O-H bonds and striae (Fig. 2f) may indicate the presence of hydrous weak minerals near the slip surfaces, in which most of the shear deformation was localized. Consequently, the locally high concentration of clay minerals next to and on the slip surfaces could make the weakening of the fault possible.

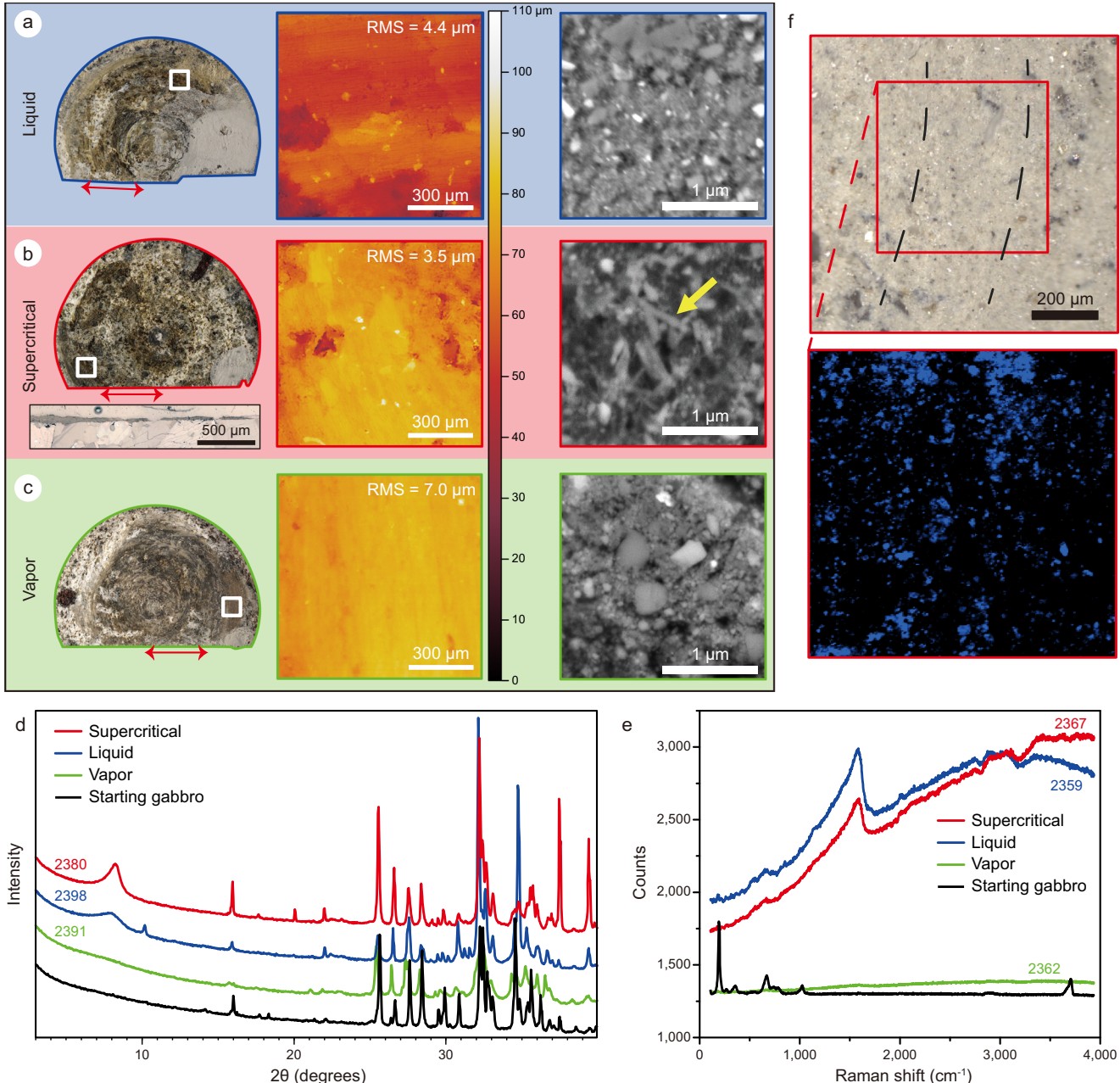

**Fig. 2 | Microstructural and mineralogical analysis on post-experimental samples.** The gabbro cylinders were recovered after shearing under hydrothermal conditions. **a–c** Data for tests performed at slip velocity $V = 10\,\mu m/s$ with water in liquid (LHV2398, shading in blue color), supercritical (LHV2380, shading in red color) and vapor (LHV2391, shading in green color) states. The left panel shows photographs of the slip surface. The middle panel shows the digital elevation model of slip surface derived from optical profilometer over area of $1 \times 1\,mm^2$ (square in the left panel, Supplementary Fig. 4). The right panel shows scanning electron microscope images perpendicular to the slip zone along the profile marked by red arrow in the left panel. In **b**, a panoramic view of the slip zone is shown in the inset in left panel, and the newly-formed clay-like minerals are indicated by yellow arrow (LHV2380). **d** X-ray powder diffraction data. The peak at ~8° suggests the formation of clay minerals and phyllosilicates in experiments LHV2398 (liquid state water) and LHV2380 (supercritical state water) (see also Supplementary Fig. 5). **e** Micro-Raman spectra data for starting sample (black in color) and for experiments performed at $V = 1\,mm/s$ with water in liquid (LHV2359, blue in color), supercritical (LHV2367, red in color) and vapor (LHV2362, green in color) states. A broad peak at a wavenumber of ~1600 $cm^{-1}$ (corresponding to H–O–H bending vibration mode) is only present in experiments with liquid and supercritical water. **f** Micro-Raman map for the experiment LHV2367 (supercritical water; scanned area is marked by red square) showing the spatial distribution of H–O–H bending bonds (in blue) in the slip zone.

Additional causes of the measured low $\mu_{ss}$ could be the chemical effects of water in the slipping zone. The micro-Raman spectra of the slip zones recovered from the experiments performed with water in liquid and supercritical states have a peak at ~1600 $cm^{-1}$ (Fig. 2e) that indicates a vibrational mode of absorbed water typical of H–O–H bending. This signal is detected only in the slip zones associated with

low measured friction, suggesting that the presence of water and this particular type of chemical bonding of absorbed water contributes to fault lubrication. In fact, the peak (~1600 $cm^{-1}$) measured in the low-frequency range (1580–1650 $cm^{-1}$) of the H–O–H bending mode corresponds to weaker intermolecular O•••H hydrogen–bonding[45], which would contribute to the decrease of the fault strength. Therefore, we

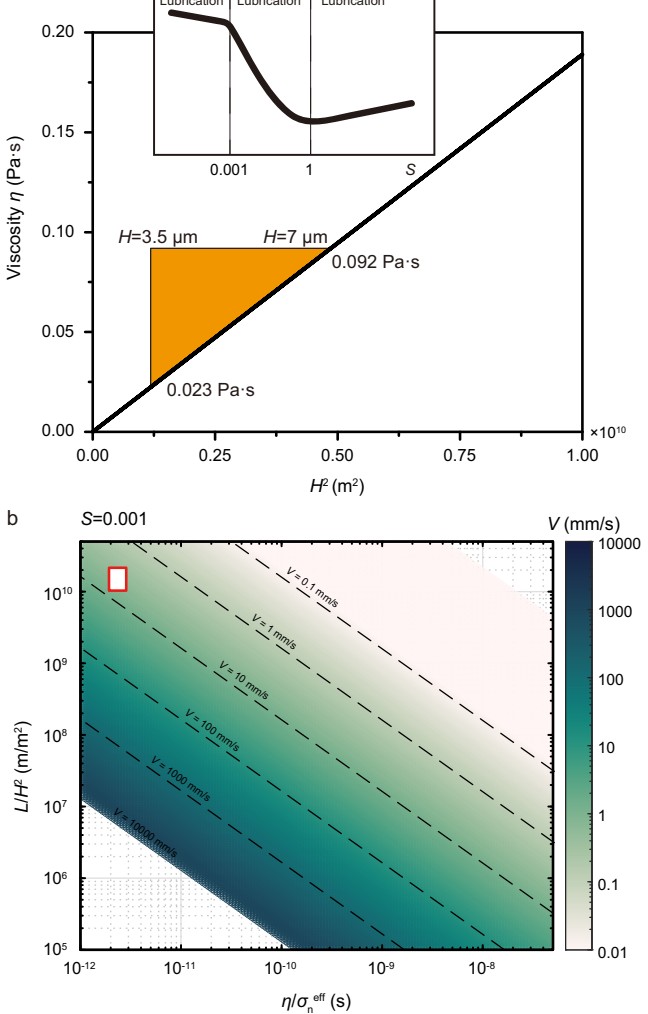

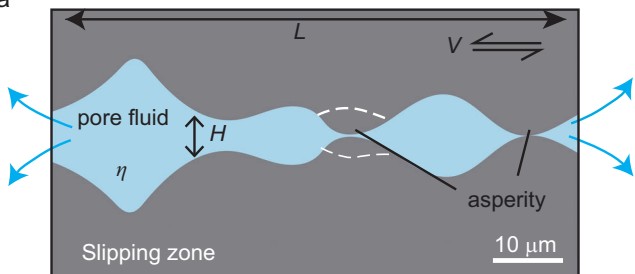

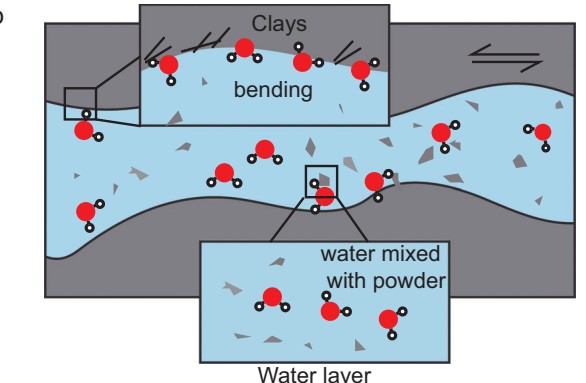

**Fig. 4 | Schematic diagram showing the weakening mechanisms operating during fault sliding in the presence of water in liquid and supercritical states. a** At the beginning of the experiment, the slip zone includes hot, pressurized water and two fault surfaces have microscopic asperities. The viscosity of the fluids ($\eta$), the height of the asperities ($H$) and the characteristic length for pressure changes ($L$) are the parameters of the Sommerfeld number. **b** During sliding, the rubbing of the surfaces produces rock powders; water-rock interaction results in the formation of clay minerals and phyllosilicates (Fig. 2d). This assemblage of slip zone traps water which has a vibrational mode (Fig. 2e) typical of H–O–H bending. This bending may result in a weakening of the intermolecular hydrogen–bonding O•••H. The water in liquid and supercritical states mixed with rock powders exhibits a significant increase in viscosity, potentially facilitating the activation of EHD mixed lubrication (Fig. 3).

**Fig. 3 | Estimate of the conditions for the activation of the EHD mixed lubrication. a** Viscosity $\eta$ vs. height of the asperities $H^2$. Inset shows the friction coefficient ($\mu$) vs. Sommerfeld number ($S$). By re-arranging equation $S = \frac{6\eta VL}{H^2 \sigma_n^{\text{eff}}}$ for $\eta = f$ ($H^2$), where $L$ is the characteristic length for pressure changes, $V$ is the slip velocity and $\sigma_n^{\text{eff}}$ is effective normal stress, the range of $\eta$ for having $S \geq 0.001$ according to the measured $H$ values (Fig. 2) spans from 0.023 to 0.092 Pa·s. **b** The multiaxis plot[50] of the Sommerfeld number for $S \geq 0.001$. The slip velocity for the activation of EHD mixed lubrication in the presence of water in vapor state is roughly 10 mm/s (red in color box). This slip velocity is consistent with the one below which the high friction coefficient is observed in experiments conducted in the presence of water in vapor state (Fig. 2f). See Fig. 4a for description of the parameters $\eta$, $H$, and $L$ of the Sommerfeld number.

propose that the hydrogen bonding between silanol (Si–O–H) trapped on the slip surface may contribute to the weakening of the fault. This possible weakening associated with the chemical bonds of absorbed water is quite different from the established weakening mechanisms. Further studies on the microstructures and dynamics bonds are required to quantify the role of absorbed water in affecting the frictional behavior of fault zones.

Considering the presence of fluid between two rough surfaces, a possible "physical" weakening mechanism could be the elastohydrodynamic (EHD) lubrication[46,47]. The adimensional Sommerfeld number (see Figs. 3a, 4a)

$$S = \frac{6\eta VL}{H^2 \sigma_n^{\text{eff}}} \qquad (1)$$

(with $\eta$ the fluid viscosity and $H$ the height of the asperities and $L$ the characteristic length for pressure changes) describes the transition, for $S \geq 0.001$, from boundary (high $\mu_{ss}$) to mixed (intermediate $\mu_{ss}$) and, for $S \geq 1$, to hydrodynamic lubrication (low $\mu_{ss}$). By re-arranging Eq. 1 for $\eta = f$ ($H^2$), we estimate the range of $\eta$ and $H$ values required to have $S \geq 0.001$ at $V = 10$ µm/s (graphic solution in Fig. 3a). Assuming that the measured RMS corresponds to $H^{46,47}$ (3.5 µm $<H<7$ µm, Fig. 2a–c), the required $\eta$ for $S \geq 0.001$ spans from $2.3 \times 10^{-2}$ to $9.2 \times 10^{-2}$ Pa s (Fig. 3a). Since the viscosity of the mixture increases significantly as the volume fraction of powder increases[47,48], these estimated viscosities can be attained when the produced rock powders mix with water in either liquid ($\eta_{\text{H2O}} = 9.0 \times 10^{-5}$ Pa·s at $T = 300$ °C, $P_f = 10$ MPa) or supercritical ($\eta_{\text{H2O}} = 4.0 \times 10^{-5}$ Pa·s at $T = 400$ °C, $P_f = 30$ MPa) states. This combination results in forming a "slurry-like" mixture or suspension, potentially facilitating the activation of mixed lubrication. Instead, the rock powders are hardly dissolved or suspended in vapor[49], therefore the viscosity of water vapor is barely affected by powders and remains very low. Consequently, the slip velocity for activation of $S \geq 0.001$ is estimated to be 9.2 mm/s[50] (Fig. 3b), which agrees with the high $\mu_{ss}$ measured at $V \leq 1$ mm/s (Fig. 1f). Indeed, the highest shortening rates are measured in the experiments performed with liquid and supercritical water (Supplementary Fig. 10), implying that the fluid expelled from the slipping zone is mixed with rock debris. This is in agreement with the presence

of trapped water in the recovered slip zones (Fig. 2e). As a consequence, we propose that a viscous "slurry-like" mixture of rock powders and water can function as a lubricant, allowing a mixed EHD lubrication process that weakens the frictional interface. In fact, this process is expected to be more efficient in the case of liquid water, since its low compressibility enables the slurry to flatten elastically the micro-roughness of the sliding surface. Such elastic flattening effects may not operate in the case of supercritical water that is more compressible, hydrodynamic lubrication can be activated through the presence of a slurry-like film in solid-solid contacts.

In this study, we show that the friction of gabbro rocks is controlled by the physical state of water. Experimental and microanalytical investigations suggest that fault weakening at low slip velocities is slip distance-dependent, probably related to the physical (mixed with powder to form viscous lubricant) and chemical (e.g., H−O−H bending) role of water together with the formation of newly-formed clay minerals (Fig. 4). The physical state of water on fault friction has significant implications for earthquake physics such as rupture nucleation, propagation and arrest. For instance, slip zone dilatancy can cause drops in pore pressure and the transition of water from liquid to vapor state and, as a result, an abrupt increase in fault friction that may arrest or delay the earthquake nucleation stage (see high-temperature slip transients in subduction zones[51]). Our work emphasizes that such a fault restrengthening process is not only caused by increased effective normal stress as previously recognized[27], but also promoted by the increase in intrinsic friction. Similarly, during rupture propagation, slip zone dilatancy along extensional jogs and step-overs may result in instantaneous transitions from liquid to vapor which may increase fault strength[52,53]. Our results provide new insights into the processes responsible for stabilizing the fault slip and arresting the rupture nucleation in natural and human-induced earthquakes.

## Methods
### Friction experiments
The experiments were performed in a Low to High-Velocity rotary shear machine, located at the State Key Laboratory of Earthquake Dynamics at Institute of Geology, China Earthquake Administration (IGCEA)[26], equipped with a dedicated hydrothermal pressure vessel. More detailed descriptions are shown in Supplementary Fig. 1. The rock used for friction experiments was gabbro named "Jinan dark green". Mineralogical composition could be found in Supplementary Tables 1, 2. The samples for experiments were machined into cylinders of 28 mm in diameter and $25 \pm 1$ mm in height. Two parallel grooves of 12 mm in length and 3 mm in width at the base of samples were carved on both sides to fit with the sample holder. Samples were jacketed with Nickel-made rings rather than Aluminum-made rings, commonly used for room-temperature experiments, to avoid rock failure during the shearing. On both upper and lower columns, graphite rings of ~20 μm diameter smaller than the internal diameter of the chamber were used as adjusters for better alignment of the columns.

The experiments were all performed under drained conditions (ISCO syringe pump maintained the system at a constant pore pressure). An effective normal stress of 10 MPa was imposed to the rock and remained constant throughout each experiment. The temperature was measured with a thermocouple located approximately at the center of the rock cylinder and ~1 mm away from the slip surface.

The experimental sequence was:

(1) pre-grinding process was conducted at 2 cm/s, under 0.5 MPa normal stress to reduce the misalignment of rock surfaces until the oscillation of axial displacement in one revolution is less than 5 μm;

(2) powders produced on the sliding surfaces were removed by using pressurized air and ethanol-wetted papers;

(3) the sample assemblage was re-mounted into the machine;

(4) the vessel was vacuumed to 2000 Pa;

(5) water was injected into the vessel and the pore pressure increased to the target value;

(6) the normal stress was loaded to the target value;

(7) the vessel was heated with the external furnace and the pore pressure was kept at a constant value (the amplitude of the pressure oscillation was less than 0.5%, see Supplementary Fig. 2a);

(8) once the desired temperature was achieved (this took 1-2 h, Stage 1, Supplementary Fig. 2a), samples were sheared at a constant velocity spanning from 1 μm/s to 0.1 m/s for a slip distance up to 3 m (slip distance, slip velocity and shear stress were determined using the methods outlined in Ref. 6) (Stage 2, Supplementary Figs. 2a, b). Normal load was measured with a load cell with a resolution of 10 N. Shear torque was recorded by the cantilever-type torque gauge consisting of a horizontal arm and a pair of force gauges with a resolution of 0.01 Nm. Axial displacement (i.e., sample plus loading column) was measured by a linear variable differential transducer (LVDT) with a resolution of 1 μm. The angular velocity and rotary angle were recorded with an encoder and a high-precision potentiometer, respectively. Pore pressure was measured by a pressure transducer (100 MPa full range with 0.01 MPa resolution) positioned between the outlet of the syringe pump and the inlet of the vessel. The temperature was measured by a K-type thermocouple (resolution 0.1 °C) placed ~ 1 mm away from the slip surface. All parameters were acquired at a frequency up to 1 kHz. When the target slip distance was achieved, we stopped the motor and turned off the furnace. The cooling process to room temperature lasted about 2 h (Stage 3, Supplementary Fig. 2a);

(9) the samples were recovered from the vessel chamber and prepared for micro-analyses.

Fault strength was presented as friction coefficient $\mu$, the ratio of measured shear stress to effective normal stress. For each experiment, the static friction coefficient ($\mu_s$) or the friction coefficient value when sliding starts on the fault, and peak friction coefficient ($\mu_p$) were obtained from the curve of friction coefficient with time (Supplementary Fig. 2). We determined the steady-state friction coefficient ($\mu_{ss}$) as the average value within a distance window of ~200 mm which covers ~4 rotations of the cylindrical samples to minimize the effects of the frictional fluctuations due to the rotary configuration (see Supplementary Table 1 and Supplementary Fig. 2). Correspondingly, the shortening (or dilatancy) rate at steady-state was calculated as the ratio of axial displacement to slip distance over the same slip distance range. To assess the effect of the physical state of water on fault friction, experiments with liquid, vapor and supercritical water were performed under otherwise identical loading conditions. Although the final slip distance is relatively small in the experiments performed at $V \le$ 0.1 mm/s, it is worth noting that, the evolution of fault friction is remarkably different depending on the physical state of water.

### Modeling of the temperature increase in the slipping zone
The temperature increase due to shear heating was estimated as first-order using a coupled Finite Element Analysis 2-Dimensional time-dependent model in Matlab®. In this model, we computed the heat source and heat dissipation in time and space. We considered a 2-D sample (28 × 25 mm, or the diameter versus height of each cylinder of gabbro) bounding the water-saturated slip zone. The initial effective normal stress was equal to the one imposed in the experiments. Fluid thermal and physical (e.g., thermal conductivity, density, compressibility) properties were taken from NIST database as a function of temperature $T_0$ and fluid pressure $P_0$ imposed during the experiments (e.g., vapor, supercritical and liquid states; see Supplementary Table 3). Two different materials were used to simulate the experimental slip zone and the wall rock (gabbro). The wall rock was

regarded as very low porous media, whereas the initial porosity of the slip zone was $\phi = 1 - A_r/A = 0.95$, where $A_r$ is the real contact area and $A$ is the nominal area of the slip surface. The slipping zone has a width of 100 µm. The thermal and hydraulic properties of the slipping zone were $k_{eff} = (1-\phi) \times k_r + \phi \times k_f$ and $\rho C_{eff} = \rho_r \times C_r \times (1-\phi) + \rho_f \times C_f \times \phi$, where the subscripts $r$ and $f$ are related to the rock and fluid properties, respectively. The fluids and rock properties were considered constant during the modeled experiments. In the model, the experimental fault is sheared at the imposed slip velocity $V(t, r)$. We assumed that all the mechanical energy is dissipated as heat[54,55] (possibly overestimate), so the heat flux $Q(r,t) = 0.5 \cdot \tau(t) \cdot V(r,t)$ is a function of time $t$ and the radial distance $r$ from the center of the sample. A Neumann boundary condition was applied to the bottom external edge (i.e., slip zone) of the model to consider the flux of heat due to shearing. On the other three external boundaries, a constant temperature $T = T_0$ as the initial experimental temperature of the two materials and a constant fluid pressure $P_f = P_0$ function of the experimental conditions (Supplementary Table 3) for the slipping zone were imposed. At the inner boundary between the slip zone and the wall rock, the continuity of the solution was granted.

### Elastohydrodynamic (EHD) lubrication
EHD was proposed to explain the reduction in friction during seismic slip when viscous fluids are present between rough surfaces of tectonic faults[46]. Following Eq. (1), the Sommerfeld number $S$ controls the transition from boundary lubrication (solid-solid contacts) to mixed lubrication (mixed contacts) to full hydrodynamic lubrication (interstitial fluid film). In the case of vapor condition ($T = 400\,°C$, $P_f = 10$ MPa) in our experiments, $\eta_{H2O} = 2.5 \times 10^{-5}$ Pa s, the rock surface at the end of the experiment has a roughness of RMS ~ 3.5 µm. The perimeter of our sample (88 mm) is considered as the characteristic length $L$. For activation of $S \geq 0.001$, the required slip velocity is roughly estimated to be higher than 10 mm/s.

### X-Ray powder diffraction
X-ray powder diffraction (XRPD) measurements were performed at the Department of Geoscience at the University of Padova using a Philips X'Pert Pro MPD diffractometer. The instrument is equipped with a long-fine-focus cobalt anode tube working at 40 kV–40 mA and a 240 mm goniometer radius that operates in the θ/θ geometry. Incident beam optics include the Bragg–BrentanoHD (BBHD) module: a wafer crystal of W/Si manufactured to improve signal-to-noise and peak-to-background ratios while maintaining a divergent beam and reducing Kα-2 and Kβ lines. Divergence slits of ¼°, antiscatter slits of 1° and Soller slits of 0.04 rad complete the incident beam setup.

Diffracted beam optics are composed of antiscatter slit of 9.1 mm aperture, Soller slits of 0.04 rad and X'Celerator Position Sensitive Detector with a 2.122° 2θ active length.

Measurements for phase identification were carried out between 3° and 85° 2θ angle, using a 0.017° step size, counting 100 s per virtual step on a spinning sample (1 revolution per second): total scan time is 1 h and 6 min. Samples were prepared using the front-loading procedure onto a Si-crystal sample holder produced to have no diffraction lines (zero-background) suitable for the small amount of material available. Oriented and Ethylene glycol saturated samples were measured between 2.5° and 42° 2θ angle, using a 0.033° step size, counting 200 s per virtual step.

Mineralogical species in bulk samples were identified using the search and match procedure implemented in PANalytical High Score Plus v.4.9.0 (Malvern Panalytical Ltd, Malvern, UK). Phyllosilicates and clay groups were recognized[29].

### Micro-Raman spectroscopy
Raman spectra were collected at the Department of Geoscience at the University of Padova, using WITec AlphaR confocal microscope with a

*XYZ* motorized stage and equipped with the 532 nm laser diode (maximum power: 60 mW) and with a 300 lines/mm grating that ensures a ~ 2 cm$^{-1}$ resolution over a spectral range from 0 to 4000 cm$^{-1}$.

Raman spectra were recorded with an integration time of 0.25/0.5 s and for each spectrum, 30 scans were accumulated. The applied power ranges from 5 to 10 mW.

### Non-contact optical profilometer
The surface morphometrics was performed at the Department of Geoscience at the University of Padova with a portable non-contact 3D optical profilometer Nanovea Jr25, equipped with an optical pen providing a vertical resolution (height repeatability) of 3.4 nm. The measurements were done on $2 \times 2$ mm$^2$ areas, using a lateral resolution (step size) of 2 µm and a scan speed of 4 mm/s (2000 Hz). The 3D maps were processed with the software Gwyddion v2.61 (Czech Metrology Institute). The final calculation (selected $1 \times 1$ mm$^2$) of the texture parameters, including those from the standard EN ISO 25178-2, was performed by a 3D analysis of the height maps including the high-pass roughness component and the low-pass waviness component together (that is, no wavelength cut-off was set).

### Field emission gun scanning electron microscope
SEM images were collected on a Tescan Solaris Field-Emission SEM at the Department of Geoscience at the University of Padova. Images have been acquired with an in-beam mid-angle backscattered detector using an accelerating voltage of 5 KeV, current of 300 pA and a working distance of 3 or 4 mm.

## Data availability
The experiment data files used in this study are available at https://zenodo.org/record/7520657.

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

## Acknowledgements

Toshi Shimamoto is thanked for the design of the hydrothermal vessel. The experimental work conducted at IGCEA was financially supported by the National Natural Science Foundation of China (grants 42174223, 41774191, and 42111530030 to L.Y., and grant U1839211 to S.M.). W.F. acknowledges the Chinese Scholarship Council (201906440130) for offering a scholarship to support his Ph.D. study. R.G. has received funding from the European Union's Horizon 2020 research and innovation program under the Marie Skłodowska-Curie grant agreement No 896346—FRICTION. G.D.T. acknowledges the ERC CoG NOFEAR (614705), the Italian Civil Protection project EXTEND and Ministero dell'Università e della Ricerca project PRIN 2022WE2JY9. Marco Favero

(XRPD), Lisa Santello (Raman), Leonardo Tauro (sample preparation), Stefano Castelli (microphotographs) and Jacopo Nava (FEG-SEM) are thanked for their technical support. Yongsheng Zhou, Giorgio Pennacchioni, Stefano Aretusini, Elena Spagnuolo, Marie Violay, André Niemeijer, Yikai Liu, Jietuo Wang, Lisong Wang, Chen Yang, Wei-Hsin Wu, Telemaco Tesei and Jianye Chen are thanked for useful discussions.

## Author contributions

W.F., L.Y., S.M., and G.D.T. conceived the study. W.F. and L.Y. performed the experiments with inputs from C.Y. and G.D.T. C.C. performed the temperature simulations. W.F. and R.G. carried out the microstructural analysis. L.G. and C. M. carried out the roughness analysis. W.F. wrote the first draft of the manuscript with inputs from L.Y. and G.D.T. All authors discussed and interpreted the results.

## Competing interests

The authors declare no competing interests.
