## [Peer Review File · Nature Communications]

Physical state of water controls friction of gabbro-built faultsREVIEWER COMMENTS

Reviewer #1 (Remarks to the Author):

The manuscript by Wei Feng and co-authors presents laboratory-derived evidence of fault weakening that appears to be controlled by the phase of water present at the frictional interface. With the help of a new hydrothermal rotary shear apparatus, the authors have probed a previously unexplored combination of controlled parameters:

1. sliding velocities spanning 5 decades, ranging from nucleation up to intermediate sub-seismic values,
2. large slip distances, and
3. water in liquid, vapor, or supercritical form.

The authors argue that previously documented mechanisms, i.e., dynamic weakening and strain localization in shear bands composed of aligned phyllosilicate minerals, are insufficient to explain the results of their experiments. Instead, based on the mechanical data from the experiments as well as extensive post-mortem analyses of the gabbro specimens, the authors propose two possible weakening mechanisms:

1. a chemistry-driven mechanism, whereby water in liquid or supercritical state weakens the gabbro interfaces to produce gouge consisting of weakly-bonded phyllosilicate minerals, and
2. a mechanical/physical mechanism, whereby the mixing of gouge particles with water produces a lubricant of appropriate viscosity to cause macroscopic weakening of the frictional interface.

A couple of limitations of the study, that, in my opinion, do not weaken its key conclusions, are:

1. The comparatively small total displacement in some of the experiments with the lowest sliding velocities, which may have prevented the system from reaching its steady-state. However, it is understandable that at such slow sliding velocities, reaching large displacements would require a significant amount of time.
2. For the modelling of temperature jumps, the assumption that all the mechanical energy is converted into heat is problematic, as there is evidence of gouge production in the experiments. In light of this fact, should the results of the modelling (Supplementary Figure 7) be treated as an "overestimation"?

The main contribution of this study is that it highlights the role of water's physical state in enabling the operation of the proposed weakening mechanisms. I believe that the explanation of the first (chemical) weakening mechanism could be improved by highlighting the contrast with already established mechanisms mentioned earlier in the discussion (also given the unfortunate fact that the produced gouge could not be fully recovered post-mortem). Overall, I recommend the publication of the manuscript with minor modifications/clarifications to address that issue as well as some points mentioned below.

== Specific comments ==

Lines 16, 17 (Abstract) and 62, 64 (Results): it seems to me that the two sentences are somewhat contradictory regarding the role of velocity in the case of water in the liquid state. According to Figure 1, panels a & b, there is certainly weakening, although a trend with velocity may be difficult to discern (especially if the μ_{ss} values for the lowest sliding velocities do not capture the true steady-state).

Line 163: Omit the dash in "a-dimensional" (or replace the entire term with "dimensionless"?)

Line 305: Replace "were" with "are"?

Line 330: replace "takes" with "took" to keep the style consist with line 336 ("lasted")?

Figure 1: It would be helpful if the colors of the markers in panels b, d, and f be kept consistent with the colors of the corresponding experiments in panels a, c, and e. Having them in a different color for each water state does not seem to serve any purpose.

Figure 4: Panel b could benefit from some light reworking. For example the background of the lower inset ("water layer") could be made opaque and light blue.

Reviewer #2 (Remarks to the Author):

This paper presents new laboratory data on frictional behavior of simulated gabbro faults over a broad range of slip velocities to meter-scale slip distances, systematically contrasting the distinct responses associated with fluid in the forms of water in vapor, liquid and supercritical states. The data are of high quality, and they fill in a critical gap in our understanding of the coupling of phase transition of the saturating fluid and its potential subtle impacts on fault strength and stability. Plausible mechanisms related to water-rock interaction and elastohydrodynamic lubrication are proposed herein to interpret the experimental data. Both methodology and results are presented in details. My primary comments are on certain technical aspects and the Discussion.

On the numerical modeling, although the exact width may not impact the final results, the authors should specify what the width of the water-saturated slip zone was in their simulations.

L161 "However, further studies are required to quantify the role of chemical bonds of absorbed water in fault slip zones" – exactly what types of studies do the authors envision that can clarify this subtle process?

L171-174 "These estimated viscosities are in the range of those attained by a mixture or suspension of rock powders with water in liquid and supercritical states...." citing references 42 & 43" – the narrative here is misleading and seems to imply that there exist directly relevant measurements on these plausible values. The wording needs to be cleaned up to explicitly provide the logical argument.

L 304 L332-333 In which lab were these experiments conducted? Although they may have been presented in earlier publications, relevant technical details on accuracy and precision of measurements including slip distance, velocity, stresses, torque, pressure and temperature should also be provided here.

Typos:

L27 "faults"

L41 "is" -> are

L46 "for" -> For

L78 "much extent"

L163 "a-dimensional" -> "adimensional" or "nondimensional"

Reviewer #3 (Remarks to the Author):

This manuscript presents results of experimental measurements of frictional properties of gabbro at high-temperature hydrothermal conditions. The authors used a confined rotary shear apparatus to investigate how friction evolves as a function of slip and slip rate, specifically focusing on the state of aqueous pore fluid. The authors conducted experiments spanning 3 possible states of interstitial water: liquid, steam, and supercritical fluid. This is important because transitions between the states may significantly affect the fluid properties, solid-fluid interactions, and frictional strength. Such transitions can occur both as a function of depth, and as a function of slip rate when coseismic heating may substantially modify the local temperature on a slip interface. The authors find that the fluid state indeed controls the frictional properties of gabbro: for liquid water, friction is both slip and velocity weakening, while for steam (vapor) and supercritical state, friction is nearly velocity-neutral (but still slip-weakening). The authors attribute such different behaviors in part to formation of clay minerals, which they verified using a post-experiment microscopy analysis. The presented results are new and warrant publication in Nature Communications. Overall, the manuscript is well written and logically organized. The experimental procedures are well documented, and the methods are state of the art. The results should be of interest to a broad readership, including, but not limited to, experimentalists, seismologists, modelers, and field geologists. I recommend that this manuscript is published, pending a moderate review to address several comments, as described below.

1) While the experimental part of the study is robust, I am less convinced by the interpretation that appeals to the elasto-hydro-dynamic lubrication (EHDL) mechanism. For this mechanism to work, the fluid has to be essentially incompressible. The latter condition applies to water in a common liquid state, but not to water vapor or supercritical fluid, both of which are highly compressible, and therefore would not be very efficient at "flattening" the slip interface by elastic

deformation. Between the three fluid stated, the EHDL would be most efficient in case of liquid water; however, the experiments show that in this case the residual dynamic strength is nearly independent of the slip rate (Fig. 1a,b). Based on these arguments, the EHDL is an unlikely mechanism responsible for the observed variations in dynamic friction.

2) The authors do not mention thermal pressurization as a possible mechanism. Is this because the slip rates are too low to cause a large enough temperature increase? Please elaborate. (The argument about fluid compressibility raised above would also apply in case of thermal pressurization.)

3) All of the presented experiments exhibit evolution of friction from "peak" to "steady state" over a characteristic distance of the order of 1 m (Fig.1 a,c,e). A similar evolution (although of smaller magnitude) was reported in rotary shear experiments on dry gabbro (e.g., www.nature.com/articles/nature11370). Could there be a common mechanism, possibly enhanced by the presence of fluid? Please discuss.

4) line 35. Instead of "at seismic deformation conditions", consider "at seismic slip rates", or "at seismogenic conditions".

5) line 78. "to a much extent" -> "to a much greater extent".

6) line 90. "After the experiments, the slip surfaces were ultra-polished and lineated" - it would help to include a figure illustrating this, perhaps in Supplementary Materials.

Response to reviewers' comments

We would like to thank all three reviewers for their constructive comments on our manuscript titled “*Physical state of water controls friction of gabbro-built faults*”. The manuscript has been modified and significantly improved by their insightful suggestions. Below, you will find our point-by-point response (blue fonts) to the reviewers' comments (black fonts). Line numbers refer to those in the revised version of the manuscript.

Reviewer #1 (Remarks to the Author):

The manuscript by Wei Feng and co-authors presents laboratory-derived evidence of fault weakening that appears to be controlled by the phase of water present at the frictional interface. With the help of a new hydrothermal rotary shear apparatus, the authors have probed a previously unexplored combination of controlled parameters:

1. sliding velocities spanning 5 decades, ranging from nucleation up to intermediate sub-seismic values,
2. large slip distances, and
3. water in liquid, vapor, or supercritical form.

The authors argue that previously documented mechanisms, i.e., dynamic weakening and strain localization in shear bands composed of aligned phyllosilicate minerals, are insufficient to explain the results of their experiments. Instead, based on the mechanical data from the experiments as well as extensive post-mortem analyses of the gabbro specimens, the authors propose two possible weakening mechanisms:

1. a chemistry-driven mechanism, whereby water in liquid or supercritical state weakens the gabbro interfaces to produce gouge consisting of weakly-bonded phyllosilicate minerals, and
2. a mechanical/physical mechanism, whereby the mixing of gouge particles with water produces a lubricant of appropriate viscosity to cause macroscopic weakening of the

frictional interface.

A couple of limitations of the study, that, in my opinion, do not weaken its key conclusions, are:

1. The comparatively small total displacement in some of the experiments with the lowest sliding velocities, which may have prevented the system from reaching its steady-state. However, it is understandable that at such slow sliding velocities, reaching large displacements would require a significant amount of time.
2. For the modelling of temperature jumps, the assumption that all the mechanical energy is converted into heat is problematic, as there is evidence of gouge production in the experiments. In light of this fact, should the results of the modelling (Supplementary Figure 7) be treated as an "overestimation"?

The main contribution of this study is that it highlights the role of water's physical state in enabling the operation of the proposed weakening mechanisms. I believe that the explanation of the first (chemical) weakening mechanism could be improved by highlighting the contrast with already established mechanisms mentioned earlier in the discussion (also given the unfortunate fact that the produced gouge could not be fully recovered post-mortem). Overall, I recommend the publication of the manuscript with minor modifications/clarifications to address that issue as well as some points mentioned below.

We are grateful to the reviewer for the positive comments on our manuscript, and also for appreciating the importance of our work. We give our response to the comments below.

The reviewer highlighted that:

“The comparatively small total displacement in some of the experiments with the lowest sliding velocities, which may have prevented the system from reaching its steady-state. However, it is understandable that at such slow sliding velocities, reaching large displacements would require a significant amount of time. ”

Indeed, as the reviewer recognized, achieving an identical large displacement of 3 m in the experiments performed at a low slip velocity (e.g., 83.3 hours for $V = 10 \mu\text{m/s}$) can be quite time-consuming. However, the primary reason for not performing these long-lasting experiments is related to safety concerns: although the cooling system of the vessel is quite efficient, extended durations of such experiments could potentially lead to hazardous overheating. We are currently working in order to reduce this hazard. In any case, the fault friction indeed did not reach a steady-state value in the experiments performed at $V \leq 0.1 \text{ mm/s}$. It is important to note that, the mechanical data collected for the shorter but identical slip displacements of 0.3 m and 1 m for $V = 0.01 \text{ mm/s}$ and 0.1 mm/s , respectively, still provide useful information about the effects of the physical states of water on fault friction at these low slip velocities. As a consequence, the relatively small displacements imposed in the low velocity experiments do not weaken the main conclusions of our study. To address this point, we now added the following statement in the Methods section (Lines 417-421):

“To assess the effect of the physical state of water on fault friction, experiments with liquid, vapor and supercritical water were performed under otherwise identical loading conditions. Although the final slip distance is relatively small in the experiments performed at $V \leq 0.1 \text{ mm/s}$, it is worth noting that, the evolution of fault friction is remarkably different depending on the physical state of water.”

Regarding the numerical modeling of temperature rise, we agree with the reviewer that our results may not be accurate if we assume all mechanical energy is converted to heat. However, this assumption has been widely used in previous studies, and justified by the good agreement between the temperature estimated with this hypothesis and the temperature measured by thermocouples at the given locations (Kitajima et al., 2010; Han et al., 2011; Yao et al., 2013; 2016; Cornelio et al., 2019). Moreover, Aretusini et al., 2021 demonstrated that, at least for a similar experimental configuration (solid rock cylinders), almost all the energy dissipated during high velocity friction experiments is converted into heat. With regard to the energy sink associated with gouge production, previous work suggested that the energy consumed by grain crushing and comminution

only accounts for a small fraction of the frictional work (0.02%~1.06 % as indicated by BET surface area measurements of quartz grains before and after high-velocity friction experiments; Togo and Shimamoto, 2012). It is clear that gouge production is a relatively small energy sink. Additionally, there are a few more factors, such as the width of the slipping zone and the heat convection of the fluid, that may affect the estimated temperature rise. To avoid adding the complex discussion above in the main text, we simply cited several references to support this assumption of heat source (ref. number 54, 55; line 438) and stated that the temperature modeling results are first-order estimates (and possibly overestimate) of the temperature increase in the slipping zone (line 423 and line).

I believe that the explanation of the first (chemical) weakening mechanism could be improved by highlighting the contrast with already established mechanisms mentioned earlier in the discussion.

Moreover, in the Discussion section about the possible “chemical” weakening mechanism, we have made slight adjustments to the sentence structure to enhance logical flow. We also highlighted the distinctiveness of the “chemical” weakening mechanism proposed, and specified that understanding the microstructure and dynamics of hydrogen bonds in future studies may be useful to clarify this “chemical” weakening mechanism (Lines 181-184).

“This possible weakening associated with the chemical bonds of absorbed water is quite different from the established weakening mechanisms. Further studies on the microstructure and dynamics of hydrogen bonds are required to quantify the role of absorbed water in affecting the frictional behavior of fault zones.”

== Specific comments ==

Lines 16, 17 (Abstract) and 62, 64 (Results): it seems to me that the two sentences are

somewhat contradictory regarding the role of velocity in the case of water in the liquid state. According to Figure 1, panels a & b, there is certainly weakening, although a trend with velocity may be difficult to discern (especially if the μ_{ss} values for the lowest sliding velocities do not capture the true steady-state).

We thank the reviewer for pointing out these unclear descriptions. In the abstract (now Lines 17-18), we aim to highlight that the slip-weakening behavior (i.e., *friction decreases with slip distance*) is independent of slip velocity, rather than the dependence of friction coefficient on the imposed slip velocity. Furthermore, in the results (now Lines 61-67), we describe that these friction parameters (i.e., μ_s , μ_p and μ_{ss} values) have a slight tendency to decrease with increasing slip velocity. They are actually the different observations. Following the comments of the reviewer, and to avoid confusing the reader, we now added additional text at lines 64 and 67:

“Regardless of the imposed slip velocity, slip-weakening behavior was observed.”

“Despite the similar trends of slip weakening, the values of μ_s , μ_p and μ_{ss} , though with some scatter, decreased with increasing slip velocity.”

Line 163: Omit the dash in "a-dimensional" (or replace the entire term with "dimensionless"?)

Change made.

Line 305: Replace "were" with "are"?

Thanks, “were” has been replaced with “are”.

Line 330: replace "takes" with "took" to keep the style consist with line 336 ("lasted")?

Thanks, it has now been corrected.

Figure 1: It would be helpful if the colors of the markers in panels b, d, and f be kept consistent with the colors of the corresponding experiments in panels a, c, and e. Having them in a different color for each water state does not seem to serve any purpose.

We thank the reviewer for the suggestions on Figure 1 and we do agree that Figure 1

would be improved if the colors of the slip velocity markers are kept consistent in the left and right panels. At the same time, we would also like to keep the consistency of “blue in color = liquid water; green in color = vapor water; red in color = supercritical water” in Figures 1 and 2 for all the plotted curves and symbols for data obtained from friction experiments and various analysis. Taking into account the reviewers’ comments and our concerns, we have decided to modify Figure 1 as shown below (see Updated Figure 1).

Updated Figure 1

Figure 4: Panel b could benefit from some light reworking. For example the background of the lower inset ("water layer") could be made opaque and light blue.

We thank the reviewer for this helpful suggestion. Changes made in Figure 4 as suggested.

Updated Figure 4.

References

- Aretusini, S. et al. Fast and Localized Temperature Measurements During Simulated Earthquakes in Carbonate Rocks. *Geophys. Res. Lett.* **48**, doi:10.1029/2020gl091856 (2021).
- Cornelio, C., Spagnuolo, E., Di Toro, G., Nielsen, S. & Violay, M. Mechanical behaviour of fluid-lubricated faults. *Nat. Commun.* **10**, 1274, doi:10.1038/s41467-019-09293-9 (2019).
- Han, R., Hirose, T., Shimamoto, T., Lee, Y. & Ando, J.-i. Granular nanoparticles lubricate faults during seismic slip. *Geology* **39**, 599-602, doi:10.1130/g31842.1 (2011).
- Kitajima, H., Chester, J. S., Chester, F. M. & Shimamoto, T. High-speed friction of disaggregated ultracataclasite in rotary shear: Characterization of frictional heating, mechanical behavior, and microstructure evolution. *J. Geophys. Res.*

Solid Earth **115**, doi:<https://doi.org/10.1029/2009JB007038> (2010).

Togo, T. & Shimamoto, T. Energy partition for grain crushing in quartz gouge during subseismic to seismic fault motion: An experimental study. *J. Struct. Geol.* **38**, 139-155, doi:<https://doi.org/10.1016/j.jsg.2011.12.014> (2012).

Yao, L., Ma, S., Niemeijer, A. R., Shimamoto, T. & Platt, J. D. Is frictional heating needed to cause dramatic weakening of nanoparticle gouge during seismic slip? Insights from friction experiments with variable thermal evolutions. *Geophys. Res. Lett.* **43**, 6852-6860, doi:<https://doi.org/10.1002/2016GL069053> (2016).

Yao, L., Shimamoto, T., Ma, S., Han, R. & Mizoguchi, K. Rapid postseismic strength recovery of Pingxi fault gouge from the Longmenshan fault system: Experiments and implications for the mechanisms of high-velocity weakening of faults. *J. Geophys. Res. Solid Earth* **118**, 4547-4563, doi:<https://doi.org/10.1002/jgrb.50308> (2013).

Reviewer #2 (Remarks to the Author):

This paper presents new laboratory data on frictional behavior of simulated gabbro faults over a broad range of slip velocities to meter-scale slip distances, systematically contrasting the distinct responses associated with fluid in the forms of water in vapor, liquid and supercritical states. The data are of high quality, and they fill in a critical gap in our understanding of the coupling of phase transition of the saturating fluid and its potential subtle impacts on fault strength and stability. Plausible mechanisms related to water-rock interaction and elasto-hydrodynamic lubrication are proposed herein to interpret the experimental data. Both methodology and results are presented in details.

We would like to express our sincere gratitude to the reviewer for his/her positive assessment and the constructive comments. We have done our best to address his/her comments as shown below.

My primary comments are on certain technical aspects and the Discussion.

On the numerical modeling, although the exact width may not impact the final results, the authors should specify what the width of the water-saturated slip zone was in their simulations.

The width of the slipping zone in our model is 100 μm .

In light of the fact that, unfortunately, the gouge layer was not fully recovered after the experiments, resulting in a thin recovered gouge layer. Based on previous experiments performed, but at room temperature, in the presence of liquid water in solid gabbroic rocks (Violay et al., 2014,2015; Mizoguchi et al., 2006), we assigned a thickness of $\sim 100 \mu\text{m}$ to the slipping zone in our model. This information is now clarified in the Methods section (Line 433).

“The slipping zone has a width of 100 μm .”

L161 “However, further studies are required to quantify the role of chemical bonds of absorbed water in fault slip zones” – exactly what types of studies do the authors envision that can clarify this subtle process?

In our opinion, infrared spectroscopy, micro-Raman spectroscopy and hyper-Raman spectroscopy (Yu et al., 2020; Seki et al., 2020) could provide rich information on the microstructure and dynamics of hydrogen bonds of water. These may help us to better understand the role of absorbed water in affecting the frictional behavior of fault zones. We have now added a few words to clarify this (Lines 182-184).

“Further studies on the microstructure and dynamics of hydrogen bonds are required to quantify the role of absorbed water in affecting the frictional behavior of fault zones.”

L171-174 “These estimated viscosities are in the range of those attained by a mixture or suspension of rock powders with water in liquid and supercritical states....” citing references 42 & 43” – the narrative here is misleading and seems to imply that there exist directly relevant measurements on these plausible values. The wording needs to be cleaned up to explicitly provide the logical argument.

Thank you for the constructive comments. We rewrote these sentences as follows (lines

194-201).

“Since the viscosity of the mixture increases significantly as the volume fraction of powder increases^{46,47}, these estimated viscosities can be attained when the produced rock powders mix with water in either liquid ($\eta_{H_2O} = 9.0 \times 10^{-5} \text{ Pa}\cdot\text{s}$ at $T = 300 \text{ }^\circ\text{C}$, $P_f = 10 \text{ MPa}$) or supercritical ($\eta_{H_2O} = 4.0 \times 10^{-5} \text{ Pa}\cdot\text{s}$ at $T = 400 \text{ }^\circ\text{C}$, $P_f = 30 \text{ MPa}$) states. This combination results in forming a “slurry-like” mixture or suspension, potentially facilitating the activation of mixed lubrication.”

L 304 L332-333 In which lab were these experiments conducted? Although they may have been presented in earlier publications, relevant technical details on accuracy and precision of measurements including slip distance, velocity, stresses, torque, pressure and temperature should also be provided here.

Thank you for pointing this out. We added the following text in the Methods section (Lines 365-366 and 396-404).

“The experiments were performed in a Low to High-Velocity rotary shear machine, located at the State Key Laboratory of Earthquake Dynamics at Institute of Geology, China Earthquake Administration (IGCEA)²⁶, equipped with a dedicated hydrothermal pressure vessel.”

“Normal load was measured with a load cell with a resolution of 10 N. Shear torque was recorded by the cantilever-type torque gauge consisting of a horizontal arm and a pair of force gauges with a resolution of 0.01 Nm. Axial displacement (i.e., sample plus loading column) was measured by a linear variable differential transducer (LVDT) with a resolution of 1 μm . The angular velocity and rotary angle were recorded with an encoder and a high-precision potentiometer, respectively. Pore pressure was measured by a pressure transducer (100 MPa full range with 0.01 MPa resolution) positioned between the outlet of the syringe pump and the inlet of the vessel. The temperature was measured by a K-type thermocouple (resolution 0.1 $^\circ\text{C}$) placed $\sim 1 \text{ mm}$ away from the slip surface.”

Typos:

L27 “faults”

Done.

L41 “is” -> are

Thanks, “is” has been replaced with “are”.

L46 “for” -> For

Change made.

L78 “much extent”

Change made.

L163 “a-dimensional” -> “adimensional” or “nondimensional”

Change made, thanks.

References

Mizoguchi, K., Hirose, T., Shimamoto, T. & Fukuyama, E. Moisture-related weakening and strengthening of a fault activated at seismic slip rates. *Geophys. Res. Lett.* **33**, doi:10.1029/2006GL026980 (2006).

Yu, C.-C. *et al.* Vibrational couplings and energy transfer pathways of water’s bending mode. *Nat. Commun.* **11**, 5977, doi:10.1038/s41467-020-19759-w (2020).

Seki, T. *et al.* The bending mode of water: A powerful probe for hydrogen bond structure of aqueous systems. *J. Phys. Chem. Lett.* **11**, 8459-8469, doi:10.1021/acs.jpcllett.0c01259 (2020).

Violay, M. *et al.* Effect of water on the frictional behavior of cohesive rocks during earthquakes. *Geology* **42**, 27-30, doi:10.1130/g34916.1 (2014).

Violay, M., Di Toro, G., Nielsen, S., Spagnuolo, E. & Burg, J. P. Thermo-mechanical pressurization of experimental faults in cohesive rocks during seismic slip. *Earth*

Reviewer #3 (Remarks to the Author):

This manuscript presents results of experimental measurements of frictional properties of gabbro at high-temperature hydrothermal conditions. The authors used a confined rotary shear apparatus to investigate how friction evolves as a function of slip and slip rate, specifically focusing on the state of aqueous pore fluid. The authors conducted experiments spanning 3 possible states of interstitial water: liquid, steam, and supercritical fluid. This is important because transitions between the states may significantly affect the fluid properties, solid-fluid interactions, and frictional strength. Such transitions can occur both as a function of depth, and as a function of slip rate when coseismic heating may substantially modify the local temperature on a slip interface. The authors find that the fluid state indeed controls the frictional properties of gabbro: for liquid water, friction is both slip and velocity weakening, while for steam (vapor) and supercritical state, friction is nearly velocity-neutral (but still slip-weakening). The authors attribute such different behaviors in part to formation of clay minerals, which they verified using a post-experiment microscopy analysis. The presented results are new and warrant publication in Nature Communications. Overall, the manuscript is well written and logically organized. The experimental procedures are well documented, and the methods are state of the art. The results should be of interest to a broad readership, including, but not limited to, experimentalists, seismologists, modelers, and field geologists. I recommend that this manuscript is published, pending a moderate review to address several comments, as described below.

We would like to express our gratitude to the reviewer for acknowledging the importance of our work. We also thank the reviewer for the constructive and insightful comments that prompted us to carefully rethink the interpretations. We have done our best efforts to respond to these comments as follows.

1) While the experimental part of the study is robust, I am less convinced by the interpretation that appeals to the elasto-hydro-dynamic lubrication (EHDL) mechanism. For this mechanism to work, the fluid has to be essentially incompressible. The latter condition applies to water in a common liquid state, but not to water vapor or supercritical fluid, both of which are highly compressible, and therefore would not be very efficient at "flattening" the slip interface by elastic deformation. Between the three fluid stated, the EHDL would be most efficient in case of liquid water; however, the experiments show that in this case the residual dynamic strength is nearly independent of the slip rate (Fig. 1a,b). Based on these arguments, the EHDL is an unlikely mechanism responsible for the observed variations in dynamic friction.

Thank you for raising this critical point. We agree with the reviewer that the compressibility of water in vapor and supercritical states is significantly higher compared to water in liquid state. However, we think that for the EHD to work the fluid flow must be incompressible, rather than the fluid being incompressible. In fluid dynamics, incompressible flow (isochoric flow) refers to a flow in which the material density is constant within a fluid parcel (Sambavaram and Sarin, 2001).

The fluid's compressibility does not determine whether the fluid flow is classified as incompressible flow. However, it is worth noting that in the literature, the term "incompressible fluid flow" is sometimes simply called "incompressible fluid" (Brodsky et al., 2001). When the fluid compressibility is acceptably small, the flow is considered to be incompressible.

The EHD lubrication theory is fundamentally built on the fluid motion governed by the Navier-Stokes equation. The incompressible fluid flow is an important assumption in the model derivation, with which the time derivative of the density can be neglected and the continuity equation reduces to (Brodsky et al., 2001):

$$\frac{\delta u}{\delta x} + \frac{\delta w}{\delta z} = 0 \quad \text{Eq.1}$$

In our experiments, when the (equivalent) friction coefficient reaches steady-state values, the density of the fluid (or the density of the mixture of water and rock

powders) remains relatively constant. So it is reasonable to consider that, regardless of the water state, the continuity equation can be validly written in the form of Eq 1.

Based on these arguments, the EHD lubrication theory is still valid and can be considered a possible fault weakening mechanism. In fact, EHD lubrication theory provides a quantitative relationship between the friction coefficient (μ) and the Sommerfeld number (S), constraining three different regimes: boundary lubrication, mixed lubrication and hydrodynamic lubrication (Williams, 2005; Brodsky et al., 2001; Cornelio et al., 2019). In our study, we propose that the decrease in the friction coefficient with slip distance occurs under the mixed lubrication regime ($S \sim 0.001$), where the normal stress is supported by both solid-solid contacts and the fluid.

We agree with the reviewer that the fluid with higher compressibility would enhance the efficiency of “flattening” the slip interface. This would facilitate an elevation in net pressure rise and result in a more pronounced EHD lubrication process.

Nonetheless, it is important to note that the mixture of rock powders and water rather than solely pure water acts as the lubricant in our experiments. The compressibility of the mixture may not exhibit significant variations in the cases of water in different physical states. Moreover, the viscosity of the fluid is likely to be more important than its compressibility in order for the underlying lubrication mechanism to work. We propose that the mixture of produced gouges with liquid and supercritical water forms a slurry-like lubricant to weaken the frictional strength, as summarized by Reviewer #1 *“mixing of gouge particles with water produces a lubricant of appropriate viscosity to cause macroscopic weakening of the frictional interface”*.

By taking into account the reviewers’ comments and our understanding of EHD, we change the text as follows:

Lines 185: *“Considering the presence of fluid between two rough surfaces, ...”*

Line 208-211: *“As a consequence, we propose that a viscous “slurry-like” mixture of rock powders and water can function as a lubricant, allowing a mixed EHD lubrication process that weakens the frictional interface. In fact, this process is*

expected to be more efficient in the case of liquid water, since its low compressibility enables the slurry to flatten elastically the micro-roughness of the sliding surface.”

In fact, figuring out the exact weakening mechanisms in these experiments is really challenging, despite our intensive micro-analytical work conducted on the recovered samples. We believe that these reproducible experimental data about the relation between the physical state of water and rock friction may require further studies to investigate the underlying weakening mechanisms.

2) The authors do not mention thermal pressurization as a possible mechanism. Is this because the slip rates are too low to cause a large enough temperature increase? Please elaborate. (The argument about fluid compressibility raised above would also apply in case of thermal pressurization.)

We thank the reviewer for reminding us to discuss the possible activation of thermal pressurization (TP). As suggested by the reviewer, the temperature rises in most of our experiments are negligible except for those performed at slip velocities of 10 mm/s (measurable) and 100 mm/s (significant; towards ~ 1000°C within the slipping zone). However, TP involves competing processes between thermal and hydraulic diffusion with several key parameters that are not well defined (e.g., slipping zone width, permeability and porosity, etc. especially how they may evolve during slip; see also discussion in Cornelio et al., 2022). In any case, we do not consider TP can be a possible weakening mechanism in our experiments for two main reasons:

1. Cornelio et al., 2022 used an optimization procedure to match the mechanical data obtained by shearing solid rock in the presence of pressurized water with the predicted shear stress (that is, the dynamic frictional strength) due to the activation of thermal pressurization. They showed that in high-velocity frictional experiments where the increase of the temperature in the slipping zone is significant (about 1200°C), other weakening mechanisms are more suitable for explaining the mechanical data (see its Supplemental Figure 3, Cornelio et al., 2022).

2. Given the imposed effective normal stress of 10 MPa, the maximum slip velocity V

= 100 mm/s is still not sufficiently high to cause TP capable of explaining the weakening observed in our experiments, as suggested by numerical modeling results of thermal pressurization shown below.

We built a finite element model with Comsol Multiphysics to estimate the pore pressure rise in the slipping zone (For a detailed model description, see Yao et al., 2023). The model's geometry is built based on the real sample assembly used in the experiments. We take temperature- and pressure-dependent properties of water into account. We assume the intrinsic friction coefficient as a constant value of 0.7. The modeling results (see Rebuttal Fig. 1) show that the observed weakening at $V \leq 10$ mm/s cannot be explained by TP as TP is negligible under these conditions. At $V=100$ mm/s, TP may contribute somewhat to the observed weakening, but is still not the dominant weakening mechanism. In summary, TP process is unlikely to be a significant weakening mechanism in our experiments, especially in the cases of $V \leq 10$ mm/s.

We have added the following text and Supplementary Figure 9 to emphasize this point (Lines 144-151).

“The weakening measured at $V= 100$ mm/s could be attributed to several established weakening mechanisms (e.g., flash heating^{36,38}, thermal pressurization^{31,39}, frictional melting⁴⁰, etc.). We assess the weakening effect of thermal pressurization as an example via numerical modeling³⁹. The results show that only at $V= 100$ mm/s pore pressure increase in the slipping zone could potentially contribute to the observed weakening (see supplementary Fig. 9). Flash heating is also likely to occur at $V \geq 100$ mm/s in accordance to previous studies^{36,38}. However, the weakening behavior observed at slip velocities ≤ 10 mm/s cannot be explained by these mechanisms.”

Rebuttal Figure 1

3) All of the presented experiments exhibit evolution of friction from "peak" to "steady state" over a characteristic distance of the order of 1 m (Fig.1 a,c,e). A similar evolution (although of smaller magnitude) was reported in rotary shear experiments on dry gabbro (e.g., www.nature.com/articles/nature11370). Could there be a common mechanism, possibly enhanced by the presence of fluid? Please discuss.

Thank the reviewer for bringing to our attention the results reported in Brown and Fialko (2012), which is relevant to our work. The authors proposed that the formation of hotspots and “melt welts” unload the rest of the slip surface in experiments performed on gabbro. They observed dynamic fault weakening at slip velocities higher than 0.06-0.2 m/s, primarily driven by the local formation of “melt welts”. We have now cited this paper in the revised manuscript (ref. number 35). However, we do not believe that the weakening mechanism operating in their experiments is responsible for the slip-weakening behavior observed in our experiments. The experiments described in Brown and Fialko (2012) were performed under room humidity conditions, and the local formation of frictional melts is the key to the weakening mechanism they proposed. Even if a similar mechanism were to operate in our experiments performed at the slip velocity of 100 mm/s (comparable to 0.06-0.2 m/s), it cannot account for the weakening observed at slip velocities of 1 to 4 orders of magnitude lower (10 μ m/s to 10 mm/s). More importantly, the complex fluid-rock interaction at elevated temperatures (300-400°C) and pore pressure (30 MPa), as in our experiments, differs significantly from the conditions in the cases of Brown and Fialko (2012). As a consequence, we believe that other mechanisms beyond the “melt welts” may operate in our experiments.

4) line 35. Instead of "at seismic deformation conditions", consider "at seismic slip rates", or "at seismogenic conditions".

Done, “seismic deformation conditions” has been replaced with “seismogenic conditions”.

5) line 78. "to a much extent" -> "to a much greater extent".

Change made as suggested.

6) line 90. "After the experiments, the slip surfaces were ultra-polished and lineated" - it would help to include a figure illustrating this, perhaps in Supplementary Materials.

We thank the reviewer for the suggestions. We have now added a figure (Supplementary Fig. 4 in the revised manuscript) to illustrate the lineated slip surface. The term “*ultra-polished*” is supported by the RMS values of the slip surfaces (3.5-7.0 μm) derived from the roughness measurements (Fig. 2a-c).

In response to the reviewer’s comment, the following figure has been added as Supplementary Figure 4 in the revised manuscript.

Updated Supplementary Figure 4. “Close up” photographs of slip surfaces recovered from the experiments performed at $V = 10 \mu\text{m/s}$. The slip surfaces are polished and *striae* are parallel to the slip vector. **a.** Experiment LHV2398, water in liquid state. **b.** Experiment LHV2380, water in supercritical state. **c.** Experiment LHV2391, water in vapor state.

References

- Sreekanth R. Sambavaram, Vivek Sarin, (2002), Parallel Computational Fluid Dynamics 2001.
- Brodsky, E. E. & Kanamori, H. Elastohydrodynamic lubrication of faults. *J. Geophys. Res. Solid Earth* **106**, 16357-16374, doi:10.1029/2001jb000430 (2001).
- Brown, K. M. & Fialko, Y. 'Melt welt' mechanism of extreme weakening of gabbro at seismic slip rates. *Nature* **488**, 638-641, doi:10.1038/nature11370 (2012).
- Cornelio, C. et al. Determination of parameters characteristic of dynamic weakening

mechanisms during seismic faulting in cohesive rocks. *J. Geophys. Res. Solid Earth* **127**, e2022JB024356, doi:<https://doi.org/10.1029/2022JB024356> (2022).

Williams J., 2005. *Engineering Tribology*. Cambridge University Press.

Yao, L., Ma, S. & Di Toro, G. Coseismic fault sealing and fluid pressurization during earthquakes. *Nature Communications* **14**, 1136, doi:[10.1038/s41467-023-36839-9](https://doi.org/10.1038/s41467-023-36839-9) (2023).

REVIEWERS' COMMENTS

Reviewer #1 (Remarks to the Author):

Dear Editor,

Regarding the revised manuscript titled "Physical state of water controls friction of gabbro-built faults" by Feng et al., I would like to let you know that, in my opinion, the authors have addressed all of the points I raised in my review in a satisfactory manner. Furthermore, I believe that the authors have presented reasonable arguments in addressing the issues raised by the other reviewers.

I remain at your disposal for any further questions.

Yours faithfully,
Reviewer #1

Reviewer #2 (Remarks to the Author):

In revising the manuscript, the authors have basically address my concerns raised in my earlier review in an adequate manner.

Reviewer #3 (Remarks to the Author):

The authors have done a good job with their revision. I agree that the viscosity of a fluid/gouge aggregate likely plays a more important role than its compressibility, however I am not convinced that compressibility of a "slurry" is controlled by compressibility of solid (i.e., gouge) particles. When the effective normal stress is primarily supported by a solid phase (which would be the case for a highly compressible fluid), the shear zone material is no longer a slurry. Also, because of the relatively short wavelengths of the "bumps" on a sample interface, the elastic flattening would require very high overpressure. The bottom line is that the hydrodynamic lubrication may well be involved, while the elastohydrodynamic lubrication is less likely.

I also agree with the authors that melting is unlikely involved in their experiments, at least at the macroscopic level. My comment was referring to the "early evolution" stage of the dry gabbro experiments (Fig. 1 in Brown & Fialko, 2012) which looks similar to the results presented by the authors. If the authors have conducted any experiments at dry conditions, it would be instructive to include (or at least describe) the respective results. I understand that their apparatus might not permit experiments at dry conditions.

The above comments are secondary to the main contributions of this manuscript, and I leave it to the authors whether to consider them any further. I congratulate the authors on a nice contribution.

Response to reviewers' comments

We would like to express our gratitude to all three reviewers for their time and constructive comments on our revised manuscript titled “*Physical state of water controls friction of gabbro-built faults*”. Below, you will find our point-by-point response (blue fonts) to the reviewers' comments (black fonts).

Reviewer #1 (Remarks to the Author):

Dear Editor,

Regarding the revised manuscript titled "Physical state of water controls friction of gabbro-built faults" by Feng et al., I would like to let you know that, in my opinion, the authors have addressed all of the points I raised in my review in a satisfactory manner. Furthermore, I believe that the authors have presented reasonable arguments in addressing the issues raised by the other reviewers.

I remain at your disposal for any further questions.

Yours faithfully,

Reviewer #1

We thank Reviewer #1 for the positive assessments.

Reviewer #2 (Remarks to the Author):

In revising the manuscript, the authors have basically address my concerns raised in my earlier review in an adequate manner.

We thank Reviewer #2 for the positive assessments.

Reviewer #3 (Remarks to the Author):

The authors have done a good job with their revision. I agree that the viscosity of a fluid/gouge aggregate likely plays a more important role than its compressibility, however I am not convinced that compressibility of a "slurry" is controlled by compressibility of solid (i.e., gouge) particles. When the effective normal stress is primarily supported by a solid phase (which would be the case for a highly compressible fluid), the shear zone material is no longer a slurry. Also, because of the relatively short wavelengths of the "bumps" on a sample interface, the elastic flattening would require very high overpressure. The bottom line is that the hydrodynamic lubrication may well be involved, while the elastohydrodynamic lubrication is less likely.

We agree with the reviewer that the compressibility of a slurry-like mixture of powders and water may not be mainly controlled by the solid phase. Therefore, if the fluid is compressible (e.g., supercritical water) the normal stress will be supported by the solid phase. In this case, the elastic effect of fluid in flattening the slip surface may be limited. Instead, the hydrodynamic lubrication may operate through the lubricated solid-solid contacts by the slurry-like film. To clarify this point according to the comments from the reviewer, we added additional discussion on Lines 201-203.

“Such elastic flattening effects may not operate in the case of supercritical water that is more compressible, hydrodynamic lubrication can be activated through the presence of a slurry-like film in solid-solid contacts.”

I also agree with the authors that melting is unlikely involved in their experiments, at least at the macroscopic level. My comment was referring to the "early evolution" stage of the dry gabbro experiments (Fig. 1 in Brown & Fialko, 2012) which looks similar to the results presented by the authors. If the authors have conducted any experiments at dry conditions, it would be instructive to include (or at least describe) the respective results. I understand that their apparatus might not permit experiments at dry conditions.

We thank the reviewer for clarifying further the comment in the 1st round review. We still believe that the results reported by Brown and Fialko, 2012 are not directly

comparable to the results shown in Figure 1 in the main text. As the reviewer acknowledged, the “melt welts” mechanism proposed in their paper is unlikely activated in our experiments. Since the reviewer is curious about the experimental results at dry conditions, we show the evolution of friction coefficient with slip for the only experiment we performed at room temperature and dry conditions in Rebuttal Figure 1. In fact, this experiment is designed for other research purposes, the experiment was stopped several times during slip. Nonetheless, it is clearly shown that no weakening in friction coefficient occurred within the initial 100 mm of slip distance at dry conditions we investigated. We apologize for not being able to provide more intuitive data at this moment.

In the main text, we simply added “under room humidity” on Line 138 and cited this relevant paper (ref. 35).

Rebuttal Figure 1 Experiment LHV2450 performed at room temperature ($\sim 25^{\circ}\text{C}$), pore pressure of 0 MPa (dry) and 10 MPa effective normal stress, at a slip velocity of 100 $\mu\text{m/s}$.

The above comments are secondary to the main contributions of this manuscript, and I leave it to the authors whether to consider them any further. I congratulate the authors on a nice contribution.

Thanks to Reviewer #3 for acknowledging the contributions of our work.